EMBO
Molecular Medicine

# Intrathecal activation of CD8+ memory T cells in IgG4-related disease of the brain parenchyma

Mirco Friedrich[1,2], Niklas Kehl[1,3], Niko Engelke[3], Josephine Kraus[3], Katharina Lindner[1,4], Philipp Münch[1], Iris Mildenberger[1,3], Christoph Groden[5], Achim Gass[3], Nima Etminan[6], Marc Fatar[3], Andreas von Deimling[7,8], David Reuss[7,8], Michael Platten[1,3,9,10] (iD) & Lukas Bunse[1,3,*] (iD)

## Abstract

IgG4-related disease (IgG4-RD) is a fibroinflammatory disorder signified by aberrant infiltration of IgG4-restricted plasma cells into a variety of organs. Clinical presentation is heterogeneous, and pathophysiological mechanisms of IgG4-RD remain elusive. There are very few cases of IgG4-RD with isolated central nervous system manifestation. By leveraging single-cell sequencing of the cerebrospinal fluid (CSF) of a patient with an inflammatory intracranial pseudotumor, we provide novel insights into the immunopathophysiology of IgG4-RD. Our data illustrate an IgG4-RD-associated polyclonal T-cell response in the CSF and an oligoclonal T-cell response in the parenchymal lesions, the latter being the result of a multifaceted cell–cell interaction between immune cell subsets and pathogenic B cells. We demonstrate that CD8+ T effector memory cells might drive and sustain autoimmunity via macrophage migration inhibitory factor (MIF)-CD74 signaling to immature B cells and CC-chemokine ligand 5 (CCL5)-mediated recruitment of cytotoxic CD4+ T cells. These findings highlight the central role of T cells in sustaining IgG4-RD and open novel avenues for targeted therapies.

**Keywords** CSF single-cell sequencing; cytotoxic T helper cell; IgG4-related disease; inflammatory pseudotumor; pathogenic B-cell
**Subject Categories** Immunology; Neuroscience

## Introduction

Immunoglobulin G4 (IgG4)-related disease (IgG4-RD) is the pathological consequence of aberrant infiltration of IgG4-restricted plasma cells into a variety of organs, most commonly the pancreas and lymph nodes (Stone *et al*, 2018; Perugino & Stone, 2020). These lesions are mostly accompanied by the excessive production of IgG4, resulting in elevated IgG4 serum levels and an increased IgG4/IgG ratio (Della-Torre *et al*, 2013). Parenchymal lesions of IgG4-RD in the central nervous system (CNS) are very rare, and the majority of these have additional systemic manifestations (Kuroda *et al*, 2019; Temmoku *et al*, 2020). The pathophysiological mechanisms of IgG4-related autoimmunity remain elusive, with many immune cell subsets described to be involved in disease progression (Baptista *et al*, 2017; Perugino & Stone, 2020).

## Results

Here, we report on a 55-year-old male patient who was admitted to our hospital with progressive blurry vision and a monocular visual acuity reduction to 1% on the right side. At the time of admission, he was not taking any immunomodulatory medication. MRI scans showed a contrast-enhancing lesion within the right orbital cavity and optic channel (Fig EV1, 05/2019). According to the patient's medical history, in previous years, the patient had had progressive vertigo and right-sided hypoacusis followed by stereotactic radiosurgery for a right temporal/petrosal contrast-enhancing lesion (Fig EV1A, 09/2014). Following radiotherapy, the patient had developed structural epilepsy, a progressive sixth nerve palsy, and headache. A follow-up MRI scan had shown a progressive infiltration of the contrast-enhancing lesion extending into the right temporal lobe (Fig EV1A, 02/2015). Glucocorticoid therapy had been initiated followed by a partial resection of the lesion and adjacent temporal lobe. The histology of the resected lesion demonstrated no signs of malignancy, but unspecific inflammation. At the time of admission, all standard and extended diagnostics remained inconclusive and

1 DKTK Clinical Cooperation Unit (CCU) Neuroimmunology and Brain Tumor Immunology, German Cancer Research Center (DKFZ), Heidelberg, Germany
2 Department of Hematology, Oncology and Rheumatology, University Hospital Heidelberg, Heidelberg, Germany
3 Department of Neurology, MCTN, Medical Faculty Mannheim, Heidelberg University, Mannheim, Germany
4 Faculty of Biosciences, Heidelberg University, Heidelberg, Germany
5 Department of Neuroradiology, Medical Faculty Mannheim, Heidelberg University, Mannheim, Germany
6 Department of Neurosurgery, Medical Faculty Mannheim, Heidelberg University, Mannheim, Germany
7 Department of Neuropathology, Heidelberg University Hospital, Heidelberg, Germany
8 DKTK CCU Neuropathology, DKFZ, Heidelberg, Germany
9 Helmholtz Institute of Translational Oncology (HI-TRON), Mainz, Germany
10 Immune Monitoring Unit, National Center for Tumor Diseases (NCT), Heidelberg, Germany
*Corresponding author. Tel: +49 6221 3858; E-mail: l.bunse@dkfz-heidelberg.de

there were no extracranial disease manifestations found. However, we observed cerebrospinal fluid (CSF) pleocytosis accompanied by an isolated intrathecal immunoglobulin (Ig) production. In parallel to an intravenous glucocorticoid re-challenge (Fig EV1B–D), we therefore performed exploratory CSF single-cell sequencing (CSF scSeq) and compared it to datasets of publicly available control and multiple sclerosis (MS) patients (Fig 1A–C) (Schafflick et al, 2020). We found a pleiotropic landscape of T cells, including naïve and effector CD8$^+$ and CD4$^+$ T cells (Fig 1A). Furthermore, the inflammatory pseudotumor (IPT) CSF exhibited a significantly increased abundance of naïve and non-switched memory B cells as compared to control and MS patient-derived CSF (Fig 1B and C). Pseudotime analysis of the patient's B-cell population revealed a trajectory from these naïve, cycling B cells to IgG4-restricted B cells (Fig 1D) (Trapnell et al, 2014).

As these findings were highly suggestive of IgG4-RD, we aimed to retrospectively validate the diagnosis in accordance with the consensus statement on the pathology of IgG4-RD (Deshpande et al, 2012): Strikingly, all resected tissues showed characteristic histological features such as dense lymphoplasmacytic infiltrate, obliterative phlebitis, storiform fibrosis, and IgG4 positivity (Fig EV2). Moreover, the IgG4/IgG ratio as determined by histology was greater than 0.4, thereby fulfilling the diagnostic criteria of IgG4-RD (Figs 1E and EV3 and EV4). Lastly, we found co-localization of astrocytes and IgG4$^+$ cells, suggestive of a hitherto undescribed primary intraparenchymal manifestation (Fig EV5A). Based on the highly suggestive CSF scSeq results and in line with the clinical course with repeated sensitivity to glucocorticoids, the increased IgG4/IgG ratio as well as characteristic pathological features, the diagnosis of primary intracerebral IgG4-RD was confirmed. Based on the high abundance of naïve B cells on an IgG4 trajectory, suggestive for a strong recruitment of B cells into the CSF, we aimed to investigate the underlying mechanism. The chemokines C-X-C motif chemokine 13 (CXCL13) and CC motif chemokine ligand 21 (CCL21) are particularly known to regulate B-cell migration into the CNS and to promote intrathecal accumulation of B cells (Kowarik et al, 2012). Specifically, CXCL13 was

suggested to play a role in the formation of ectopic lymphoid tissues within the CNS (Aloisi et al, 2008). Interestingly, CXCL13 CSF levels were remarkably higher in IPT CSF compared with control or MS patient-derived CSF (Fig 1F). CD4$^+$ central memory and CD8$^+$ effector memory T cells were the immune cell subset with the highest median expression of CXCL13 (Figs 1G and EV5B). Importantly, CD4$^+$ memory T-cell subsets identified by Seurat v4 reference mapping expressed a T helper 2 cell (Th2)-associated cytokine profile including IL-4, IL-10, and IL-21. Previous studies suggest that Th2 cells drive the class switch toward IgG4 via IL-4 signaling (Baptista et al, 2017; Akiyama et al, 2018). Consistently, naïve B cells found in our IPT CSF dataset showed increased expression of IL4R. As IL-10 is suggested to preferentially promote class switch toward IgG4 over IgE (Jeannin et al, 1998), we found expression of IL10RA on all CSF-localized B-cell subsets in conjunction with a pronounced IL-10 expression in CD4$^+$ memory T cells (Fig 1G and H).

Our data suggested that the disease is driven by B cells in the CSF that are recruited to intraparenchymal lesions to become clonally expanding plasma cells secreting IgG4 that becomes detectable in the brain parenchyma and CSF. Several studies have assessed the pathological T-cell response in IgG4-RD, with a focus on Th2 cells (Zen et al, 2007; Tanaka et al, 2012; Müller et al, 2013; Heeringa et al, 2018) and, more recently, CD4$^+$ cytotoxic T lymphocytes (CD4$^+$ CTL) (Mattoo et al, 2016; Maehara et al, 2017) and PD-1$^{hi}$CXCR5$^-$ peripheral T helper (Tph)-like cells (Rao et al, 2017; Kamekura et al, 2018). Interestingly, we found an increased abundance of CD4$^+$ CTL in IPT CSF compared with control and MS patient-derived CSF (Fig 2A), while 10% of CD4$^+$ T cells in IPT CSF were Tph-like cells (Fig EV5C and D). However, CSF T-cell repertoire was polyclonal (Fig 2B), arguing against recruitment of antigen-specific T cells. Most CD4$^+$ T cells demonstrated expression of TCF7 consistent with a dysfunctional state, while most CD8$^+$ T cells exhibited a phenotype reflective of cytotoxic activity, expressing granzyme and granulysin (Fig 2C) (Li et al, 2019). We therefore aimed to further characterize the functional interactions between pathogenic B cells and highly

---

**Figure 1.  Comparative cerebrospinal fluid single-cell profiling of an inflammatory pseudotumor.**

A   Uniform Manifold Approximation and Projection (UMAP) of sequenced single cells from inflammatory pseudotumor (IPT) CSF ($n = 1$ sample, $n = 4{,}324$ single cells, left), control CSF ($n = 6$ samples, $n = 15{,}467$ single cells, middle), and multiple sclerosis (MS) patient-derived CSF ($n = 6$, $n = 18{,}412$ cells, right). Indicated immune cell subsets as identified by Seurat v4 reference (ref) mapping.

B   Stacked bar chart of relative B-cell abundances as identified by Seurat v4 reference mapping in inflammatory pseudotumor (IPT) CSF, control CSF, and MS patient-derived CSF.

C   Circle plots representing the relative abundance of B-cell subsets as identified by Seurat v4 reference mapping in inflammatory pseudotumor CSF, control CSF, and MS patient-derived CSF.

D   Pseudotime analysis of intrathecal B-cell subsets in inflammatory pseudotumor CSF with the naïve B-cell cluster, identified by canonical markers, as root node. Percentage of cycling naïve B cells and IgG4 B cells depicted in pie charts.

E   Retrospective immunohistochemistry DAB staining of IgG4 and IgG on archival temporal lobe resection tissue. As suggested by the consensus statement on the pathology of IgG4-RD (Deshpande et al, 2012), three 40x fields with the highest number of IgG4$^+$ and IgG$^+$ cells were selected, counted, and averaged within these fields. Cell counts as indicated.

F   C-X-C motif chemokine 13 (CXCL13) concentrations measured by ELISA from inflammatory pseudotumor (IPT) CSF, control CSF, and MS patient-derived CSF. Individual values, mean $\pm$ SEM; $n = 4$ experiment repeats with technical replicates.

G   Stacked bar chart depicting mean relative expression levels of T helper cell-associated cytokines in T-cell subsets as identified by Seurat v4 reference mapping in inflammatory pseudotumor CSF as in (A). TEM, T effector memory; TCM, T central memory; Treg, regulatory T cell.

H   Violin plot depicting relative expression levels of IL4R (left) and IL10RA (right) in B-cell subsets as identified by Seurat v4 reference mapping in inflammatory pseudotumor CSF.

Data information: (B, C) Cell subsets as indicated by the legend on the right.

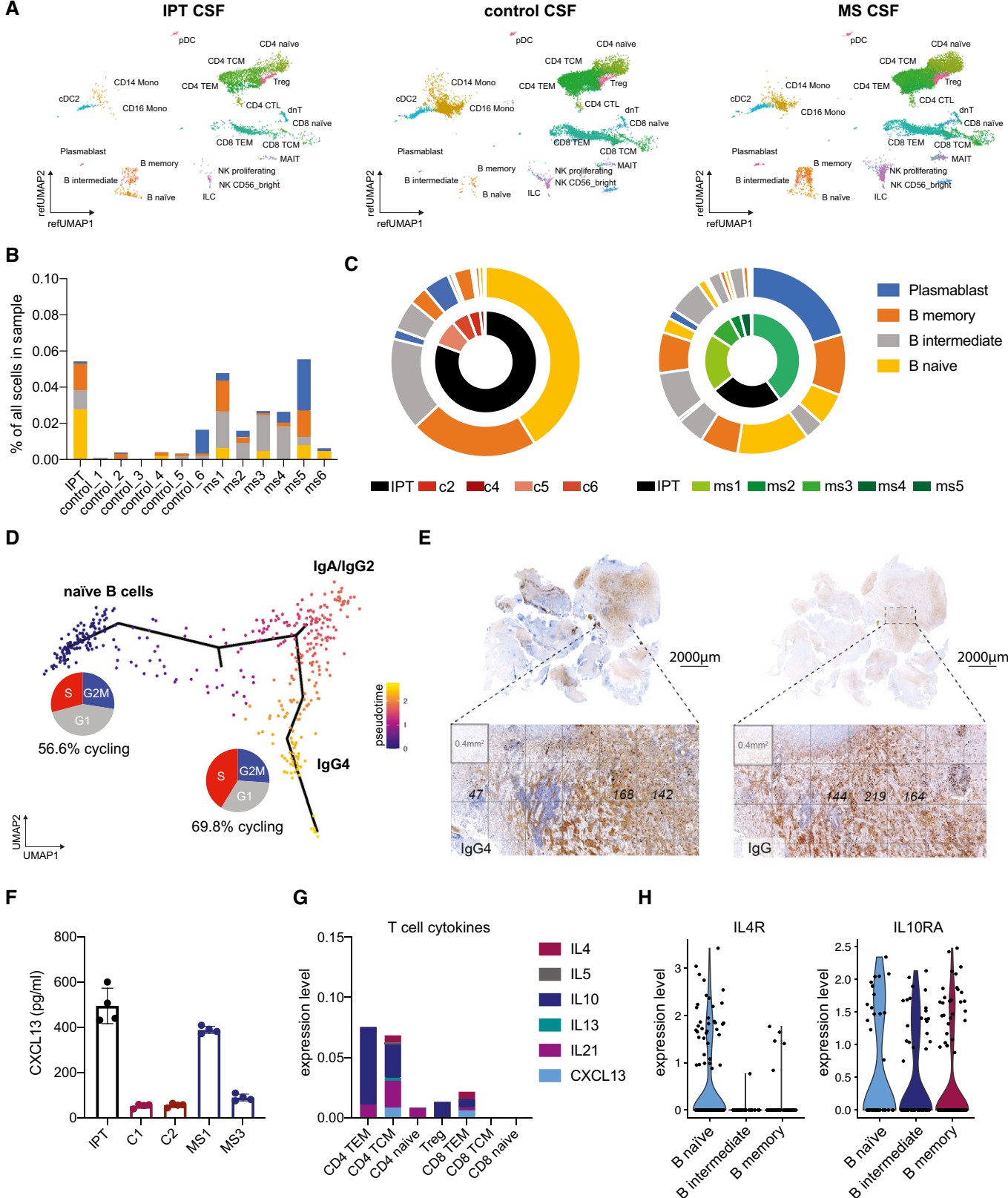

**Figure 1.**

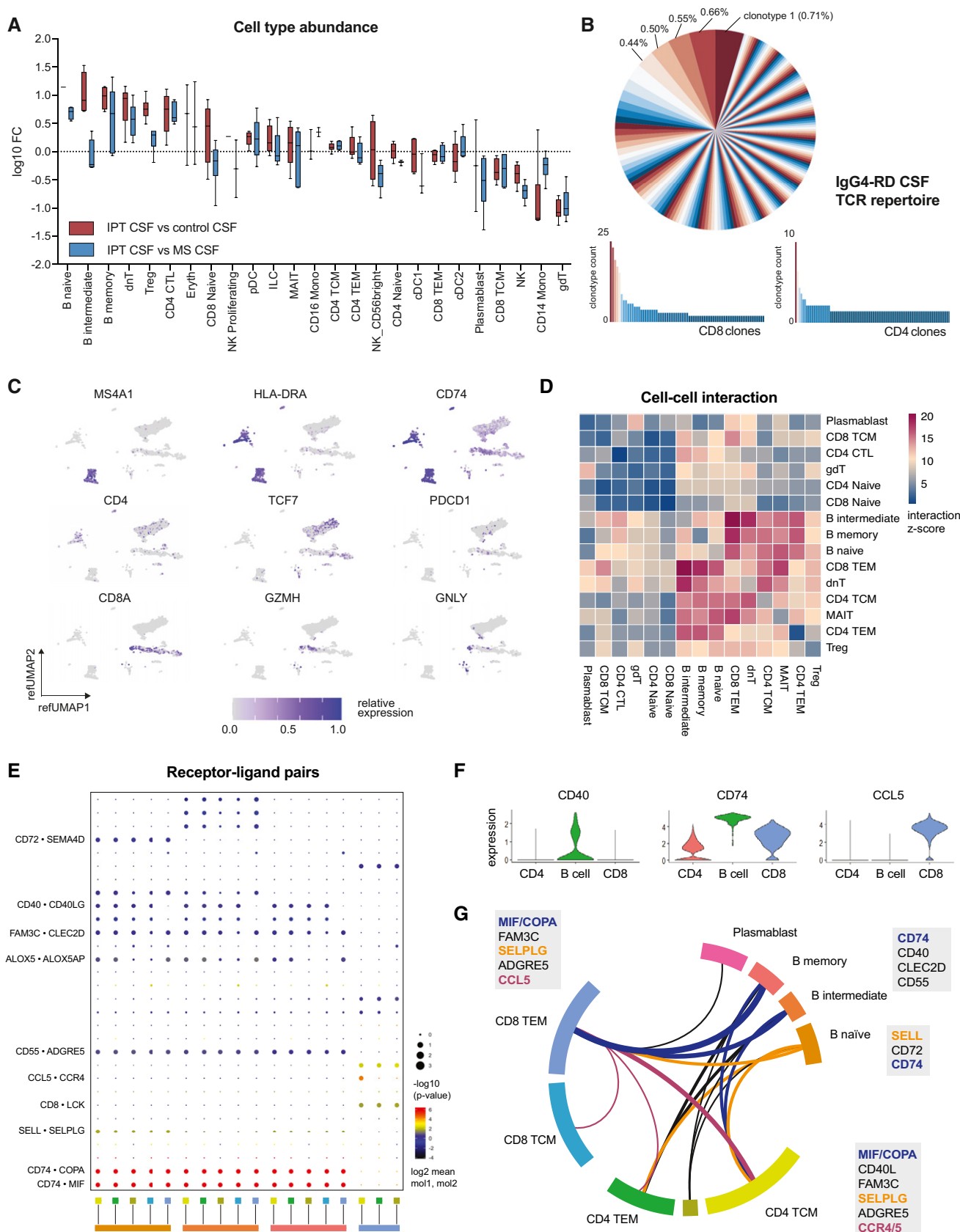

**Figure 2.**

◄

**Figure 2.  CNS manifestation of IgG4-RD with distinct cytotoxic T-cell–B-cell interactions.**

A   Relative abundances of cell types identified by Seurat v4 reference mapping as in Fig 1A in IPT CSF compared with control or MS patient-derived CSF. Boxplot depicting 25th–75th percentiles with median shown as central band and whiskers extending from minimum to maximum values. FC, fractional difference.

B   TCR repertoire of IPT CSF as analyzed by single-cell VDJ sequencing. Top, all sequenced and paired TCR clonotypes shown, clonotypes ordered counterclockwise according to abundance. Bottom, clonotype frequency stratified by T-cell subset.

C   Feature plot of UMAP of single cells from pseudotumor CSF shown in Fig 1A, depicting cell-wise representations of indicated transcripts. Relative expression shown.

D   Heatmap of cell–cell interaction analysis depicting top predicted interactions based on receptor-ligand co-expression and reference-based cell subsets. Interaction z-score shown.

E   Dot plot representation of top 30 receptor–ligand interactions based on molecule co-expression and reference-based cell subsets. Mean relative expression of both interaction partners (dot color) and interaction *P*-value (dot size) shown. *P*-values are derived from one-sided permutation tests and refer to the enrichment of the interacting ligand–receptor pair in each of the interacting pairs of cell types.

F   Relative expression of *CD40, CD74,* and *CCL5* on indicated combined meta-clusters classified by reference-based cell identification as shown in Fig 1A.

G   Circos plot representation of highlighted cell–cell interactions in IPT CSF between indicated cell subsets. Colored receptor–ligand interaction pairs from (E-F).

abundant T-cell clusters. Strikingly, we found the strongest predicted cell–cell interactions to be between pathogenic naïve, intermediate and memory B cells and CD8+ T effector memory cells (CD8 TEM) as well as a direct T-cell–T-cell crosstalk between CD8 TEM and CD4+ memory cells, while plasmablasts did not show potent intercellular interactions (Fig 2D). Analysis of co-expressed interaction partner molecules revealed that—in addition to canonical mediators of B-cell maturation, such as *CD40— CD40LG* and *ALOX5—ALOX5AP* (Nagashima *et al*, 2011)—the most highly co-expressed molecules were *CD74* on B cells and its ligand macrophage migration inhibitory factor (*MIF*) on CD8+ and CD4+ T cells, which has recently been described as B-cell chemokine that might be responsible for the migration of pathogenic B cells to IgG4-RD manifestation sites as well as their aberrant proliferation (Shi *et al*, 2006; Klasen *et al*, 2014, 2018; Della-Torre *et al*, 2020) (Fig 2E–G). Lastly, we found signals of T-cell–T-cell crosstalk via CC-chemokine ligand 5 (*CCL5*) on CD8 TEM binding to C-C chemokine receptor type 4/5 (CCR4/5)-expressing T cells that might facilitate recruitment of helper and cytotoxic CD4+ T cells that have been shown to mediate inflammation in IgG4-RD (Fig 2E–G) (Mattoo *et al*, 2016; Mattoo *et al*, 2017; Perugino *et al*, 2021).

We hypothesized that IgG4-plasma cell maturation mediated by dysfunctional T helper cells is followed by T-cell–T-cell crosstalk and a clonal cytotoxic T-cell response as effector arm of IgG4-related autoimmunity. We here propose a novel pivotal role for CD8 TEM in driving and sustaining IgG4-RD via distinct cell–cell interactions and chemoattraction of pathogenic B cells and cytotoxic CD4+ T cells. However, we observed a polyclonal TCR repertoire in the CSF in both T-cell compartments (Fig 2B). We therefore extracted FFPE-derived DNA from all available IgG4-RD lesions (2015 temporal lobe parenchyma, 2018 cavernous sinus, and 2019 optic nerve) and performed targeted TCR beta immune repertoire sequencing (TCRB-Seq). Interestingly, we found an oligoclonal TCR repertoire in all parenchymal lesions compared with the polyclonal repertoire in the CSF (Fig 3A). Repertoire distribution analysis indicated that hyperexpanded clones dominated especially the early and latest lesions (Fig 3B). When adjusting clonotype diversity for differences in library sizes across samples by rarefaction analysis, T-cell clonality consistently increased over time (2015 < 2018 < 2019; Fig 3C). TCR sequence overlap between lesions revealed greatest overlap between 2015 and 2018, followed by 2015 and 2019 as measured by Morisita's overlap index (Fig

3D). A total of 17 clonotypes were shared between all sites and time points of disease manifestation (Fig 3E), resulting in a number of predominant TCRB amino acid motifs over the course of the disease (Fig 3F). Strikingly, longitudinal tracking of the most abundant clonotypes revealed dominance of two clones at first resection (2015: CSARVDYNEQFF: 49.27%, CASSQEYSPYEQYF: 38.33%). While productive frequency of clone CSARVDYNEQFF decreased over time (2018: 9.12%, 2019: 0.07%), clone CASSQEYSPYEQYF hyperexpanded, resulting in an almost completely monoclonal disease at re-recurrence in 2019 (83.47% productive frequency; Fig 3G and H). Taken together, these findings support the notion of a dynamic, but highly clonal T-cell response as effector arm of IgG4-related autoimmunity (Perugino & Stone, 2020; Perugino *et al*, 2021).

Treatment paradigms for IgG4-RD have been extrapolated primarily from observational studies of glucocorticoids in type 1 (IgG4-related) autoimmune pancreatitis (Ghazale *et al*, 2008; Kamisawa *et al*, 2009). Following diagnosis, the here-described patient thus received high-dose glucocorticoid and consequently improved clinically (Fig EV1B). Intrathecal protein and IgG levels declined, while serum inflammation markers and immunoglobulin levels remained stable, further highlighting the restriction of active disease to the CNS (Fig EV1C and D). Similar to published casuistic reports, IgG4 concentration in the CSF was elevated (Fig EV4D), but additional studies regarding CSF IgG4 quantification and IgG4 indices for diagnostic purposes are required (Della-Torre *et al*, 2012; Della-Torre *et al*, 2014).

## Discussion

In summary, in this very rare case of primary cerebral IgG4-RD with preceding standard diagnostic workup remaining inconclusive, exploratory single-cell sequencing allowed for cell-subtype specific diagnosis and further insights into the pathogenesis of this disease. Based on our data, it is tempting to speculate that recurrent IgG4-RD is driven by an oligoclonal to clonal T-cell response. Further research is needed to determine the dynamics of clonal T-cell autoimmunity and to identify potential disease-driving epitopes in IgG4-RD. On a broader level, leveraging high-throughput single-cell technologies and detailed bioinformatic analyses may not only guide clinicians in the diagnosis of rare autoimmune disorders but also targeted treatments.

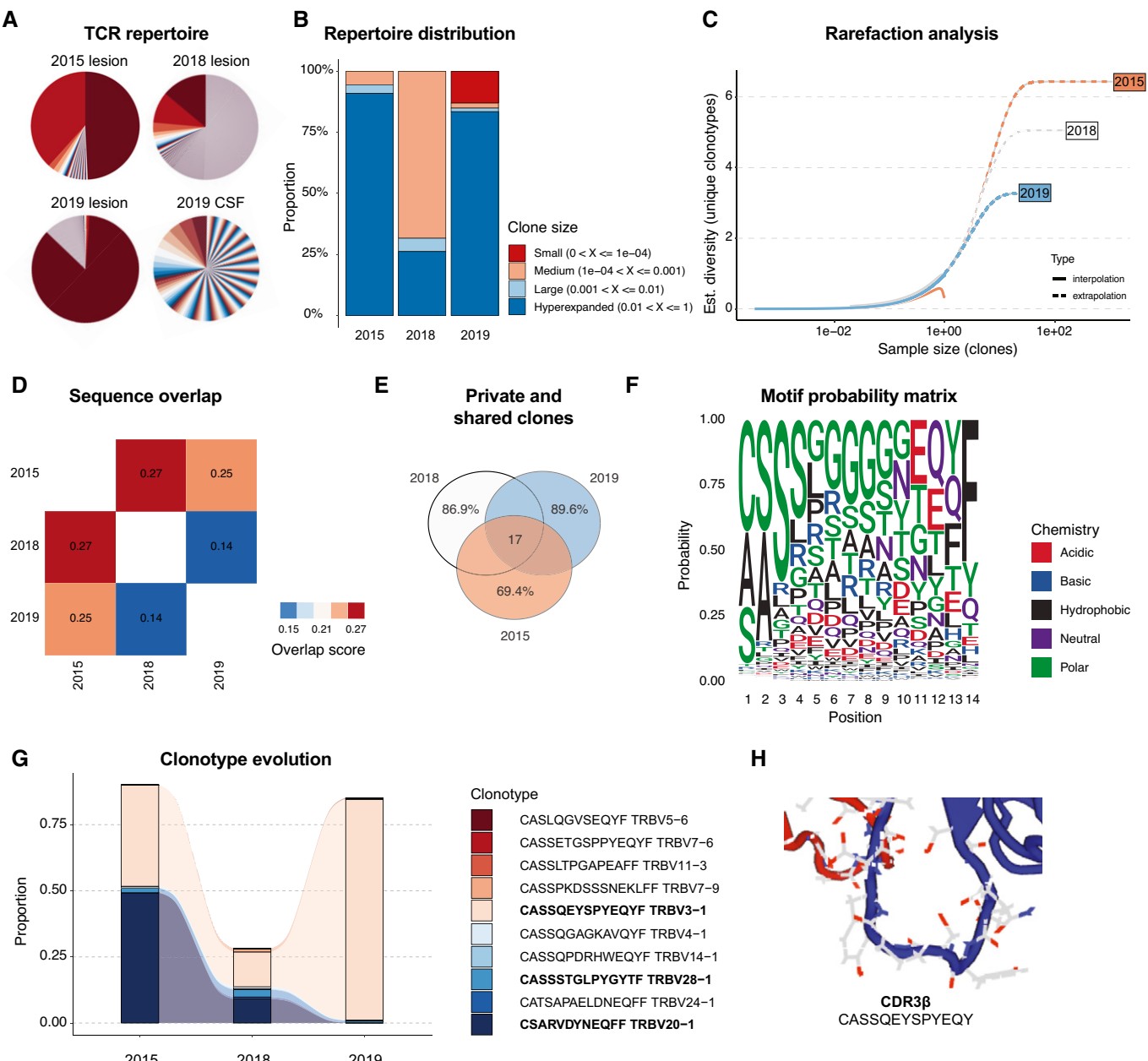

Figure 3. Clonal evolution of the TCR repertoire in IgG4-RD revealed by longitudinal VDJ- sequencing of disease manifestations.

A  TCR repertoire clonality in inflammatory pseudotumor lesions (2015–2019) and 2019 CSF as analyzed by VDJ sequencing. Clonotype abundance represented by angular diameter.
B  TCR repertoire distribution assessed as clonal space homeostasis, i.e., proportion of the repertoire occupied by the clones of a given size.
C  Rarefaction analysis indicating estimated diversity of clonotype richness in inflammatory pseudotumor lesions (2015–2019). Sample size indicated on the x-axis.
D  TCR sequence overlap between inflammatory pseudotumor lesions (2015–2019) measured by Morisita's overlap index.
E  Shared (absolute number) and unique (proportion) clonotypes between inflammatory pseudotumor lesions (2015–2019).
F  Motif probability matrix depicting occurrences of amino acids in each position of all computed CDR3 beta chains.
G  Longitudinal tracking of most abundant clonotypes in inflammatory pseudotumor lesions (2015–2019). CDR3 amino acid sequence and v-chain usage for each clonotype annotated in the legend. Private clones are limited to each time point. Shared clones are found at all time points.
H  Automated CDR3 beta chain high-resolution modeling of dominant T-cell clone CASSQEYSPYEQY.

# Materials and Methods

### Ethical approval for human samples

Written informed consent was obtained by the patient prior to this study conformed to the principles set out in the WMA Declaration of Helsinki and in the Department of Health and Human Services Belmont Report. Ethical approval for the isolation of cerebrospinal fluid and single-cell analysis was obtained from the Mannheim Medical Faculty Ethics Committee (Reference numbers 2017-589N-MA / 2019-643N-MA).

### Single-cell RNA and TCR sequencing, transcript quantification and analysis

Patient CSF cells were sequenced in multiplexed libraries on an Illumina NovaSeq sequencing system at a depth of $\sim$130,000–200,000 reads per cell. MS and control datasets were obtained from GEO with the accession code GSM4104122. Libraries with a total of 38,203 cells (Control: 15,467; MS: 18,412; patient: 4,324) after quality control were integrated and analyzed using Seurat v4 (Stuart *et al*, 2019). Data analysis was performed using the Seurat v4 workflow. Patient and reference datasets were mapped onto the multimodal PBMC reference dataset (Hao *et al*, 2021) using FindTransferAnchors() and MapQuery() with default settings. Pseudotime Analysis was performed using the Monocle package v3 (Trapnell *et al*, 2014). The CellRanger output was converted into a cell_data_set object filtering out cells with less than 200 reads per cell, corresponding to the argument umi_cutoff = 200, using the load_cellranger_data function. The cell_data_set object was preprocessed using the standard PCA method computing 100 principal components, corresponding to dims = 100 in the preprocess_cds function. Dimensionality reduction was performed using the default reduce_dimension() function. For clustering, the community detection technique was used, corresponding to the cluster_cells() function with the resolution set to 0.013. The top_markers function was used with default parameters to identify cluster-specific markers, corresponding to the argument group_cells_by = "cluster". The trajectory graphs were fitted for each partition applying the learn_graph() function. Using the choose_cells() function, a subset was generated on the cluster consisting of B cells, identified by canonical markers. Pseudotime analysis was carried out on this B cell subset, using the order_cells() function setting the naïve B-cell cluster, identified by canonical markers, as root node. Receptor–ligand interaction analyses were performed using the CellPhoneDB package (Efremova *et al*, 2020).

### TCRB deep sequencing

Genomic DNA was isolated from FFPE tissues using QIAamp DNA FFPE Tissue Kit (cat. no. 56404 QIAGEN) and Deparaffinization Solution (cat. no. 19093, QIAGEN) as per the manufacturer's instructions. TCR beta chain (TCRB) deep sequencing was performed to detect rearranged TCRβ gene sequences using hsTCRB Kit (Adaptive Biotechnologies) according to the manufacturer's protocol. The prepared library was sequenced on an Illumina MiSeq by the Genomics & Proteomics Core Facility, German Cancer Research Center (DKFZ). Data processing (demultiplexing, trimming, gene mapping) was done using the Adaptive Biotechnologies proprietary platform

as previously described (Platten *et al*, 2021). ImmunoSEQ data were exported, supplemented with metadata and analyzed with R using the immunarch 0.6.6 package infrastructure. Repertoire overlap was calculated using Morisita's overlap index. Estimation of repertoire diversity was performed using the repDiversity function. The.method parameter was adjusted to rarefaction analysis. Longitudinal clonotype tracking was calculated with the.trackClonotypes function using the following settings: Value list("2015", "10") of the.which argument. Value "aa+v" of the.col argument, so that the function takes both CDR3 amino acid sequences and V gene segments of the most abundant clonotypes. K-mer and sequence motif analysis and visualization were performed using the default getKmers() filter settings to exclude all non-coding sequences before counting the k-mer statistics. The resulting amino acid position frequency matrices (PFM) were used for motif visualization. Automated high-resolution modeling of CDR3B sequence was performed using TCRmodel (Gowthaman & Pierce, 2018).

### Immunoglobulin and cytokine ELISA

Supernatants of IgG4-RD CSF and control CSF from patients without any resulting neurological diagnosis were transferred to cytokine-specific antibody-coated or uncoated ultra-low binding 96-well plates (Corning), respectively, and immunoglobulin and cytokine ELISA based on horseradish-peroxidase were performed according to the manufacturer's instructions (Thermo Fisher, R&D Systems). Development process was stopped with 1 M $H_2SO_4$, and optical density (OD) was measured at 570 and 450 nm. Cytokine concentrations were calculated based on OD [450nm] – OD [570 nm] according to parallel serial dilutions of cytokine standards included in the respective ELISA kit. ELISA detection was used for the following human immunoglobulins and cytokines: IgG4 (IgG4 Human Uncoated ELISA Kit with Plates, #88-50590-22, Thermo Fisher), total IgG (IgG (Total) Human Uncoated ELISA Kit with Plates, #88-50550-22, Thermo Fisher), CXCL13 (Human CXCL13/BLC/BCA-1 Quantikine ELISA Kit, #DCX130, R&D Systems).

### Immunofluorescence staining

For immunofluorescence stainings, formalin-fixed paraffin-embedded (FFPE) tissue was used. After deparaffination and rehydration, antigen retrieval was performed by boiling 10 μm sections in 10 mM citric acid buffer, pH 6.0. Sections were covered with blocking solution (10% goat plasma, 2% BSA in PBS), followed by incubation with different primary antibodies overnight at 4°C: anti-GFAP (1:150 dilution; 2E1.E9; BioLegend) and anti-IgG4 (1:200 dilution; RM120; dianova). The secondary antibodies (Goat anti-Mouse IgG (H + L) Highly Cross-Adsorbed Secondary Antibody, Alexa Fluor Plus 488, Goat anti-Rabbit IgG (H + L) Highly Cross-Adsorbed Secondary Antibody, Alexa Fluor Plus 546, Thermo Fisher) were applied at 1 μg/ml to the sections in blocking buffer for 1h at RT. Tile scans (20×) and higher-magnification images (40×) were acquired using a Carl Zeiss Cell Observer HS fluorescence microscope.

### Immunohistochemistry staining of IgG4 and IgG

Immunohistochemistry was conducted on 3 μm thick FFPE tissue sections mounted on StarFrost Advanced Adhesive slides

**The paper explained**

**Problem**
IgG4-related disease (IgG4-RD) is an autoimmune disorder signified by infiltration of pathological plasma cells into a variety of organs. Clinical symptoms are diverse, and the underlying mechanisms that lead to IgG4-RD remain elusive. There are very few cases of IgG4-RD with isolated central nervous system manifestation, and treatment is unspecific and often not very successful.

**Results**
This paper aims at shedding new light into potential molecular mechanisms of cell-to-cell communication in IgG4-related disease. The authors propose the idea that the abnormal immune response in IgG4-RD is driven by single T-cell clonotypes.

**Impact**
This paper demonstrates the potential of single-cell profiling technologies to support clinicians in the diagnosis of rare autoimmune disorders. Future studies might incorporate results and hypotheses of this paper to develop new causal treatments against IgG4-RD.

(Engelbrecht, Kassel, Germany) followed by drying at 80°C for 15 min. Immunohistochemistry was performed on a BenchMark Ultra immunostainer (Ventana Medical Systems, Tucson, USA). Slides were pretreated with Cell Conditioning Solution CC1 (Ventana Medical Systems) for 32 min at room temperature. For DAB staining of IgG4 and IgG, anti-human-IgG4 (clone MRQ-44, Ventana/Roche, ready-to-use dilution) and anti-human-IgG (clone A57H, Dako, 1:200 dilution) were used as primary antibodies. Primary antibodies were incubated at 37°C for 32 min, followed by Ventana standard signal amplification, UltraWash, counter-staining with one drop of hematoxylin for 4 min, and one drop of bluing reagent for 4 min. UltraView Universal DAB Detection Kit (Ventana Medical Systems) was used for visualization. Tile scans (20×) and higher-magnification images (40×) were acquired using a Carl Zeiss Cell Observer HS fluorescence microscope.

### Semiquantitative assessment of IgG4/IgG ratio

DAP-stained FFPE tissue slides were scanned using the Zeiss Axioscan™ slidescanner. For each resection tissue, areas with the highest IgG4 staining were analyzed in accordance with the consensus statement on the pathology of IgG4-RD (Deshpande *et al*, 2012). One high-power field (HPF) was defined as 0.4 mm$^2$. Counting of IgG$^+$ and IgG4$^+$ cells was done semi-automated using the CellCount plugin of the Fiji image analysis software. The number of IgG4$^+$ and IgG$^+$ cells from 3 areas of each slide were averaged for calculation of the IgG4/IgG ratio. Quantification results were validated by automatic quantification as per default DAB parameters of QuPath 0.2.3 software.

### Data visualization

Tabular data from single-cell sequencing analyses above were processed using the tidyverse suite of packages [https://CRAN.R-project.org/package=tidyverse] and visualized in the R programming environment using the ggplot2 package. Data from all other analyses were visualized using GraphPad Prism 9.0. Figures were produced using Adobe Illustrator 2021.

### Statistics (unless otherwise mentioned)

Data are represented as individual values or as median $\pm$ SD, as indicated. Group sizes (n) and applied statistical tests are indicated in figure legends. Significance was assessed by either unpaired *t*-test analysis, paired *t*-test analysis, or two-way ANOVA analysis with Tukey's post hoc testing as indicated in figure legends. Statistics were calculated using GraphPad Prism 9.0. Due to the nature of this study, sample size determination was not applicable, as all available samples were included in this study. All cells passing QC, IPT CSF ($n = 4,324$ cells), control CSF ($n = 15,467$ cells), and MS patient-derived CSF ($n = 18,412$ cells), were included in downstream analyses on a single-cell basis in a similar procedure to other exploratory neuroinflammation studies. For functional experiments, CSF samples were blinded to the experiment performer.

## Data and materials availability

Gene expression data that support the findings of this study have been deposited in the Gene Expression Omnibus repository (E-MTAB-10479). T-cell receptor deep sequencing data have been deposited in the Adaptive Biotechnologies immuneACCESS® database (https://doi.org/10.21417/MF2021EMBOMM). All additional datasets generated or analyzed during this study are included in this published article (and its supplementary information files). Imaging source data for this manuscript can be found via URL: https://doi.org/10.6084/m9.figshare.14522397.v1.

**Expanded View** for this article is available online.

### Acknowledgements
We acknowledge the support of the DKFZ Light Microscopy Facility and the DKFZ Genomics and Proteomics Core Facility. We acknowledge the support by the Flow Cytometry Core Facility at the German Cancer Research Center. We thank T. Bunse for critically reading the manuscript. This work was supported by grants from Deutsche Forschungsgemeinschaft (DFG, German Research Foundation) – Project-ID 404521405, SFB 1389 - UNITE Glioblastoma TPB03, Swiss Bridge Foundation, Else Kröner-Fresenius Foundation (2019_EKMS.49), the Medical Faculty Mannheim at Heidelberg University ("Anerkennung von Spitzenleistung" and Seed program), and the epigenetics@dkfz program to L.B; from the Helmholtz Gemeinschaft, Zukunftsthema "Immunology and Infection" (ZT0027-WP03), the Dr. Rolf M. Schwiete Foundation (07/2017), the Sonderförderlinie "Neuroinflammation" of the Ministry of Science of Baden Württemberg, the DFG, German Research Foundation – Project-ID 404521405, SFB 1389 - UNITE Glioblastoma TPB01; Project-ID 259332240, RTG2099-P14, and the Deutsche Krebshilfe (project 70113515) to M.P. M.F. is member of the MD/PhD program at Heidelberg University. M.F. received fellowships by the Heidelberg Biosciences International Graduate School (HBIGS), the Konrad-Adenauer Foundation, the German Academic Exchange Service (DAAD), the German Academic Scholarship Foundation and the Excellence Initiative of the German Council of Science and Humanities and the German Research Foundation (DFG). K.L. is supported by the Helmholtz International Graduate School (HIGS). N.K. is supported by an MD thesis stipend of the Research Training Group (RTG) 2099 funded by the

DFG, Project-ID 259332240. Open Access funding enabled and organized by Projekt DEAL.

## Author contributions

Study design, experiments, scRNA-seq, VDJ-seq, and experimental data analysis and interpretation, and writing the manuscript: MF scRNA-seq data analysis and experiments: NK Lumbar punctures and blood sampling for diagnostics and translational analyses: NE, JK, and IM Single-cell sequencing: KL Immunofluorescence tissue staining: PM. MR imaging data interpretation and provision: CG and AG Study design and data interpretation: MFa and NEt Diagnostic tissue staining: AvD and DR Study conceptualization, data interpretation, and writing the manuscript: MP and LB.

## Conflict of interest

The authors declare that they have no conflict of interest.

## For more information

- Information for patients and caregivers: https://www.rheumatology.org/I-Am-A/Patient-Caregiver/Diseases-Conditions/IgG4-Related-Disease-IgG4-RD
- Research group website: https://www.dkfz.de/en/neuroimmunologie/index.php

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
