## [Review Process File · EMBO Molecular Medicine]

Intrathecal activation of CD8+ memory T cells in IgG4-related disease of the brain parenchyma

Mirco Friedrich, Niklas Kehl, Niko Engelke, Josephine Kraus, Katharina Lindner, Philipp Münch, Iris Mildenerger, Christoph Groden, Achim Gass, Nima Etminan, Marc Fatar, Andreas von Deimling, David Reuss, Michael Platten, and Lukas Bunse

DOI: [10.15252/emmm.202113953](https://doi.org/10.15252/emmm.202113953)

Corresponding author(s): [Lukas Bunse \(l.bunse@dkfz.de\)](mailto:l.bunse@dkfz.de)

Review Timeline:

Submission Date:	13th Jan 21
Editorial Decision:	10th Feb 21
Revision Received:	5th May 21
Editorial Decision:	26th May 21
Revision Received:	3rd Jun 21
Accepted:	4th Jun 21

Editor: *Zeljko Durdevic*

Transaction Report:

10th Feb 2021

Dear Dr. Bunse,

Thank you for the submission of your manuscript to EMBO Molecular Medicine. We have now received feedback from the three reviewers who agreed to evaluate your manuscript. As you will see from the reports below, the referees acknowledge the interest of the study but also raise serious concerns that should be addressed in a major revision.

Addressing the reviewers' concerns in full will be necessary for further considering the manuscript in our journal, and acceptance of the manuscript will entail a second round of review. EMBO Molecular Medicine encourages a single round of revision only and therefore, acceptance or rejection of the manuscript will depend on the completeness of your responses included in the next, final version of the manuscript. For this reason, and to save you from any frustrations in the end, I would strongly advise against returning an incomplete revision.

We realize that the current situation is exceptional on the account of the COVID-19/SARS-CoV-2 pandemic. Therefore, please let us know if you need more than three months to revise the manuscript.

I look forward to receiving your revised manuscript.

Yours sincerely,

Zeljko Durdevic

***** Reviewer's comments *****

Referee #1 (Comments on Novelty/Model System for Author):

This is an interesting mechanistic case report on a rare manifestation of a rare disease. I have no additional advices on the model used

Referee #1 (Remarks for Author):

This is an interesting mechanistic case report on a rare manifestation of a rare disease. The authors have elegantly applied cutting edge techniques to a rare disease and, in their hands, this was essential to achieve a final diagnosis of IgG4-RD. Yet, I have major structural and also minor

comments:

MAJOR:

I have hard time believing that while the patient was waiting for a diagnosis and a treatment for his orbital lesion, the authors had time to go through scSEQ and all the analysis performed before realizing this was IgG4RD. I therefore suggest to extensively revise the structure of the manuscript and stress the only novelty of the paper, namely a detailed CSF mechanistic study on a rare IgG4RD manifestation which is unprecedented in literature.

A diagnosis of IgG4-RD could have been in fact achieved easily in two alternative ways, either by performing a proper histologic analysis of the brain parenchyma at presentation or by measuring IgG4 in the CSF with the IgG4 Index.

This extremely complicated diagnostic process was not necessary to me and might raise perplexities on physician's clinical ability in rationally approaching a clinical case. In this sense also please consider changing the title. A possible alternative could be: "Intrathecal activation of CD8+ memory T cells in IgG4-RD of the brain parenchyma"

In addition this is not the first case of isolated brain IgG4RD involvement (see for instance: PMID: 31470167). I would not insist too much, therefore, on the clinical novelty but rather on the mechanistic implications.

MINOR

METHODS

1- please better specify where the 6 controls come from? what disease? how was normal CSF obtained?

RESULTS

1- what was CSF concentration of IgG4? and the CSF IgG4/IgG ratio? was it calculated? what about the blood brain barrier permeability indices? Consider citing PMID: 24289956 and PMID: 22393144. Adding a table with the general laboratory, serological and CSF features of the patient would be appropriate

2- I cannot see supplementary Figures attached so I cannot comment on those data.

3- When showing histological pictures, the authors should clearly highlight storiform fibrosis and obliterative phlebitis. This is, in fact a rare manifestation and all available histological informations should be provided to substantiate the finding. In addition, fibroblasts are not expected to populate the brain parenchyma as traction due to tissue scarring would stretch surrounding healthy white matter with collateral unwanted damage. Previous reports of parenchymal involvement do not show fibrosis indeed. I would expect that brain parenchyma would heal with gliosis rather than fibrosis. So please provide all possible detailed images to fully convince the readers.

4- "which has recently been described as B cell chemokine responsible for the migration of pathogenic B cells to IgG4-RD manifestation sites as well as their aberrant proliferation¹⁵⁻¹⁷" Please rephrase as references 15-17 are not dealing with IgG4RD. Indeed, the author should cite PMID: 31319101 and PMID: 32485263 where chemokines for T cell and B cell attraction (including CCL5) are discussed and T cell subsets implicated in IgG4RD are first described in detail, further reinforcing the findings reported in the present manuscript.

5- If the orbital lesion responded to steroids, the authors should provide MRI pictures of the improvement.

Referee #2 (Remarks for Author):

Friedrich, et al describe a unique case of IgG4-related disease manifesting with isolated involvement of the central nervous system. Due to the rarity of this occurrence, the diagnosis was initially not arrived upon through routine histology but subsequently suggested by single cell analysis of cells obtained from the patient's cerebrospinal fluid. Through standard single cell transcriptional analysis, the authors describe subsets of B and T lymphocytes at the site of disease, and suggest potentially important mechanisms of cell-cell interaction that may help explain the various immune cell types implicated in the pathogenesis of this disease. The data and manuscript is clearly and concisely presented and provides potential novel insights into our evolving understanding of IgG4-RD. I offer the following constructive comments:

1. It is mentioned that the patient received glucocorticoids prior to resection of the brain lesion but the timing of glucocorticoid treatment to the collection of the CSF specimen used for subsequent analyses is not clear. If the patient had been receiving glucocorticoids at the time of CSF sampling, it would be important for the authors to discuss how this therapy could have altered the overall findings in terms of cellular subsets and transcriptional profiles.
2. The patient's CSF scSeq analysis revealed a preponderance of naïve and unswitched memory B cells in the CSF fluid, presumably reflective of the parenchymal brain lesion. It would be helpful if the authors can offer an explanation for this finding. Classically considering the adaptive immune response, CXCR5 expressing naïve B cells are clustered in the B cell follicles of secondary lymphoid organs by the local expression of CXCL13. Did the scRNA seq analysis reveal a CXCL13 expressing cell population to explain this accumulation of naïve B cells in the CSF?
3. Plasmablasts have been exhaustively reported to be clonally expanded and to secrete self-reactive Ig in the context of IgG4-RD. Yet, the CSF analysis from this single subject essentially showed an absence of plasmablasts in the CSF but many IgG4 expressing cells (presumably plasmablasts and plasma cells) in the parenchymal lesion. How do you explain the lack of plasmablasts in the CSF?
4. "As these findings were highly indicative of IgG4-RD" - I would soften this sentence to "suggestive of IgG4-RD" as the primary suggestion is with the pseudotime analysis and IgG4 enrichment among the cells was the only clue to the diagnosis of IgG4-RD. This wording is used twice.
5. I am struggling a bit with making sense of Figures 1d and 1e. In Figure 1d, the vast majority of B cells analyzed in the CSF had a naïve or unswitched memory phenotype. However, in Figure 1e, the naïve contribution appears to be much smaller, no unswitched memory B cells are included in the pseudotime analysis, and the contribution of switched memory B cells appears much greater. Can the authors clarify this discrepancy?
6. Figure 1f, there appears to be a significant degree of background staining from both the IgG4 and IgG stains, which is something we often encounter in our clinical experience at least with IgG most commonly. According to some of the provided quantifications, it's not clear how true positive events were distinguished from non-specific background signal and I worry about misrepresentation of the true numbers of IgG4+ cells. Can the authors please clarify?
7. "Previous studies suggest that T follicular helper cells drive the class switch towards IgG4 via Interleukin-4 signaling." By earlier in vitro studies, it has been most strongly suggested that the combination of IL4 and IL10 is required for class-switching to IgG4. Did the authors observe a signal for IL10R transcription among the analyzed naïve B cells?

8. "Interestingly, CD4+ T cell subsets demonstrated expression of PDCD1 and TCF7 consistent with a dysfunctional state." Follicular helper T cells have been exhaustively reported in the context of IgG4-RD expanded both in the blood and in the tissues. Did the authors observe co-transcription of PD1 and CXCR5 in the same cells and if so, were these cells also expressing relevant cytokines such as IL4, IL10, IL21? PD1 expressing CD4+ T cells have also more recently been described that are CXCR5- and termed peripheral helper T cells (TPH) by Deepak Rao and colleagues in the context of rheumatoid arthritis. Were the PD1 expressing CD4+ T cells observed here more fitting of this TPH phenotype, capable of helping B cells through CD40L but lacking CXCR5 expression?

9. CD4+CTLs have been described to be the most dominant CD4 T cell type at the site of disease as referenced in this manuscript, yet their contribution among the cells analyzed in the CSF is quite small. Can the authors discuss this discrepancy with the published tissue quantitative literature?

10. The CD4+CTL response reported in IgG4-RD has consistently shown marked clonal expansion. More recently, expanded CD8+CTLs in the blood of patients with IgG4-RD were also found to be highly clonally expanded (Perugino, Kaneko, Maehara, et al, 2021). However, here the authors find "a polyclonal TCR repertoire in both T cell compartments as determined by single-cell VDJ profiling." With all suggested B-T interactions, presumed antigen presentation and the reported literature on clonal expansion, how do the authors explain the diverse TCR repertoire observed here?

11. The functional interaction analysis is very interesting. "direct T cell-T cell crosstalk between CD8 TEM and CD4+ memory cells" and "we found evidence of direct T cell - T cell crosstalk via CCL5 on CD8 TEM binding to C-C chemokine receptor type 4/5 (CCR4/5)-expressing T cells that might facilitate recruitment of helper and cytotoxic CD4+ T cells that have been implied to mediate inflammation in IgG4-RD" - CCL5 made by the CD8s may have landed on its receptor on some other cell that wasn't among the cells subjected to scRNAseq. This is also location dependent. A lot of these are speculative assertions and should be presented as such. I would avoid using such declarative words like "direct" as this interaction is rather only suggested through indirect analyses.

12. "We therefore propose a novel pivotal role for CD8 TEM in driving and sustaining IgG4-RD via distinct cell-cell interactions and chemoattraction of pathogenic B cells and cytotoxic CD4+ T cells." This is a very interesting idea and one absolutely worth investigating further. I do want to bring two references to the authors attention. One, the work by Perugino, Kaneko, Maehara et al published in JACI, mentioned above, the provides transcriptional phenotyping of CD4+CTLs and CD8+CTLs from the circulating cells from patients with IgG4-RD. CCL5 and CCR5 were two transcriptional signatures noted in the bulk effector CD4+CTL data. The 2nd, by Della-Torre et al, also published in JACI, described a potential direct role for B cells in driving fibrosis in this disease and CCL4 and CCL5 were explored extensively, both found to be secreted by pathologic B cells at the sites of disease and suggested as potential mediators to orchestrate the immune cell infiltrate in this disease. These ideas of cytokine mediated communication between relevant cell types in IgG4-RD has been a recurrent finding and these additional citations bolster the authors suggestions about the importance of these axes in this disease.

Referee #3 (Remarks for Author):

Major

Parenchymal lesions of IgG4-RD in the central nervous system (CNS) are rare and the authors described a case of an inflammatory intracranial pseudotumor and reported single-cell sequencing of patient cerebrospinal fluid (CSF). This case is rare and informative in the field of CNS lesions in IgG4-RD. However, there are some questions to be resolved.

Minor specific comments

- 1) To make a correct diagnosis of IgG4-RD, histopathological findings are the most important. However, interobserver concordance among pathologists in histopathological diagnosis is not necessarily satisfied. In the text, the authors mentioned characteristic histological features such as dense lymphoplasmacytic infiltrate, obliterative phlebitis and storiform fibrosis, but not full pictures lacking obliterative phlebitis and storiform fibrosis. Full pictures should be mentioned.
- 2) It has been established that IgG4-RD is TH2 dominated immune response, but not mentioned in anywhere results. This should be mentioned or discussed.

Point by Point Response

Referee #1

This is an interesting mechanistic case report on a rare manifestation of a rare disease. I have no additional advices on the model used.

We thank the referee for this comment.

This is an interesting mechanistic case report on a rare manifestation of a rare disease. The authors have elegantly applied cutting edge techniques to a rare disease and, in their hands, this was essential to achieve a final diagnosis of IgG4-RD. Yet, I have major structural and also minor comments:

MAJOR:

I have hard time believing that while the patient was waiting for a diagnosis and a treatment for his orbital lesion, the authors had time to go through scSEQ and all the analysis performed before realizing this was IgG4RD. I therefore suggest to extensively revise the structure of the manuscript and stress the only novelty of the paper, namely a detailed CSF mechanistic study on a rare IgG4RD manifestation which is unprecedented in literature. A diagnosis of IgG4-RD could have been in fact achieved easily in two alternative ways, either by performing a proper histologic analysis of the brain parenchyma at presentation or by measuring IgG4 in the CSF with the IgG4Index. This extremely complicated diagnostic process was not necessary to me and might raise perplexities on physician's clinical ability in rationally approaching a clinical case. In this sense also please consider changing the title. A possible alternative could be: "Intrathecal activation of CD8+ memory T cells in IgG4-RD of the brain parenchyma". In addition this is not the first case of isolated brain IgG4RD involvement (see for instance: PMID: 31470167). I would not insist too much, therefore, on the clinical novelty but rather on the mechanistic implications.

We thank the reviewer for this very constructive major comment. Moreover, we appreciate the suggestion of an alternative title and changed the title in the revised manuscript accordingly. Indeed, in parallel to the CSF single cell seq, the patient received a probationary intravenous treatment with glucocorticoids which resulted in visual acuity improvement (to a greater extent than moderate radiographic response (new EV Figure 1). Previous external gamma knife radiosurgery (2014) and the preceding differential diagnoses of a radionecrosis and neurosarcoidosis hindered the immediate diagnosis of an IgG4-RD. Applying our high throughput TCR discovery platform for personalized immune receptor immunotherapy, we were indeed able to apply our bioinformatic pipelines in a reasonable window of opportunity. However, we completely agree with the reviewer that this patient-related aspect is rather specific and therefore focused more intensively on the mechanistic aspects as suggested.

MINOR

METHODS

1- please better specify where the 6 controls come from? what disease? how was normal CSF obtained?

As healthy controls for bioinformatics analyses, we used single cell sequencing datasets from patients with idiopathic intracranial hypertension (IIH) as reported in Schafflick et al., Nature Communications, 2020. Biobanked healthy control CSF for immunoglobulin and CXCL13 quantification was obtained from the Mannheim Medical Faculty biobank (Ethics reference numbers 2017-589N-MA / 2019-643N-MA).

RESULTS

1- what was CSF concentration of IgG4? and the CSF IgG4/IgG ratio? was it calculated? what about the blood brain barrier permeability indices? Consider citing PMID: 24289956 and PMID: 22393144. Adding a table with the general laboratory, serological and CSF features of the patient would be appropriate

We have now included all available information including IgG4 CSF and serum concentration in the new EV Figure 1 and 4. We have now included the suggested references in our discussion.

2- I cannot see supplementary Figures attached so I cannot comment on those data.

We apologize that these figures were not accessible. We have substantially revised our manuscript and performed additional experiments now included in the new supplementary (EV) as well as in main figures.

3- When showing histological pictures, the authors should clearly highlight storiform fibrosis and obliterative phlebitis. This is, in fact a rare manifestation and all available histological informations should be provided to substantiate the finding. In addition, fibroblasts are not expected to populate the brain parenchyma as traction due to tissue scarring would stretch surrounding healthy white matter with collateral unwanted damage. Previous reports of parenchymal involvement do not show fibrosis indeed. I would expect that brain parenchyma would heal with gliosis rather than fibrosis. So please provide all possible detailed images to fully convince the readers.

We thank the referee for this very important comment and have now included IgG4-RD characteristic histological features including a detailed annotation in EV Figure 2. Indeed, in our casuistic report, both pachymeningeal as well as intraparenchymal manifestation were observed, explaining the co-occurrence of gliosis and fibrosis. Full images can be publicly accessed via the following accession code: <https://doi.org/10.6084/m9.figshare.14522397.v1>.

4- "which has recently been described as B cell chemokine responsible for the migration of pathogenic B cells to IgG4-RD manifestation sites as well as their aberrant proliferation¹⁵⁻¹⁷" Please rephrase as references 15-17 are not dealing with IgG4RD. Indeed, the author should cite PMID: 31319101 and PMID: 32485263 where chemokines for T cell and B cell attraction (including CCL5) are discussed and T cell subsets implicated in IgG4RD are first described in detail, further reinforcing the findings reported in the present manuscript.

We are now adequately citing the suggested references and rephrased the section. Indeed, we are enthusiastic about the recent external evidence on CCL5-dependent recruitment of CD4 T cells in IgG4-RD which is suggestive for (one) shared pathomechanism within the CNS but also in peripheral manifestations.

5- If the orbital lesion responded to steroids, the authors should provide MRI pictures of the improvement.

Visual acuity was improved; however, only moderate radiographic response was observed (new EV Figure 1).

Referee #2

Friedrich, et al describe a unique case of IgG4-related disease manifesting with isolated involvement of the central nervous system. Due to the rarity of this occurrence, the diagnosis was initially not arrived upon through routine histology but subsequently suggested by single cell analysis of cells obtained from the patient's cerebrospinal fluid. Through standard single cell transcriptional analysis, the authors describe subsets of B and T lymphocytes at the site of disease, and suggest potentially important mechanisms of cell-cell interaction that may help explain the various immune cell types implicated in the pathogenesis of this disease. The data and manuscript is clearly and concisely presented and provides potential novel insights into our evolving understanding of IgG4-RD. I offer the following constructive comments:

We thank the referee for this very positive feedback.

1. It is mentioned that the patient received glucocorticoids prior to resection of the brain lesion but the timing of glucocorticoid treatment to the collection of the CSF specimen used for subsequent analyses is not clear. If the patient had been receiving glucocorticoids at the time of CSF sampling, it would be important for the authors to discuss how this therapy could have altered the overall findings in terms of cellular subsets and transcriptional profiles.

This is a very important remark. We have now clarified in the text that at the time of CSF sampling (admission to our hospital), the patient was not taking any immunomodulatory drugs. Last glucocorticoid treatment was >1 year prior to admission.

2. The patient's CSF scSeq analysis revealed a preponderance of naïve and unswitched memory B cells in the CSF fluid, presumably reflective of the parenchymal brain lesion. It would be helpful if the authors can offer an explanation for this finding. Classically considering the adaptive immune response, CXCR5 expressing naïve B cells are clustered in the B cell follicles of secondary lymphoid organs by the local expression of CXCL13. Did the scRNA seq analysis reveal a CXCL13 expressing cell population to explain this accumulation of naïve B cells in the CSF?

This is a very interesting pathomechanistical question. Based on the fact that we found evidence of IgG4-RD by the assessment of expression profiles of B cells (trajectories) in the CSF we first hypothesized that the CSF is indeed reflective of the parenchymal brain lesion. Because of the intralesional and CSF abundance of T cells, we comparatively assessed the T cell receptor repertoires of the IgG4-RD lesions and the CSF. Whereas the CSF showed a rather polyclonal repertoire, TCR repertoires of the lesions were truly clonal. Therefore, we currently conclude that the CSF is "IgG4-RD-associated" but not "IgG4-RD-lesion reflective". In line with the referee's hypothesis, CXCL13 levels were indeed elevated in the patient's CSF (in comparison to control and MS CSF samples). Based on our single cell expression CSF dataset, CXCL13 expression counts were predominantly found in CD4/CD8 TCM but overall low compared to other cytokines/chemokines (new Figure 1g).

3. Plasmablasts have been exhaustively reported to be clonally expanded and to secrete self-reactive Ig in the context of IgG4-RD. Yet, the CSF analysis from this single subject essentially showed an absence of plasmablasts in the CSF but many IgG4 expressing cells (presumably plasmablasts and plasma cells) in the parenchymal lesion. How do you explain the lack of plasmablasts in the CSF?

This is another very important and interesting pathomechanistical question. In principle, plasmablasts and antibody secreting cells (ASC) migrate into ectopic follicular-like formations within a parenchymal lesion. Moreover, direct ASC migration as well as IgG4 diffusion into the brain parenchyma have been reported in inflammatory conditions. In MS,

germline sequences of CSF B cell clones suggest that naïve B cells may transit the blood–brain barrier / blood-CSF barrier to populate meningeal germinal center-like formations (Blauth, *Front. Immunol.*, 2015), hence, we interpret this finding as a migratory snap shot with only few plasmablasts that are “IgG4-RD-lesion reflective”.

4. "As these findings were highly indicative of IgG4-RD" - I would soften this sentence to "suggestive of IgG4-RD" as the primary suggestion is with the pseudotime analysis and IgG4 enrichment among the cells was the only clue to the diagnosis of IgG4-RD. This wording is used twice.

We agree and have rephrased the sections accordingly.

5. I am struggling a bit with making sense of Figures 1d and 1e. In Figure 1d, the vast majority of B cells analyzed in the CSF had a naïve or unswitched memory phenotype. However, in Figure 1e, the naïve contribution appears to be much smaller, no unswitched memory B cells are included in the pseudotime analysis, and the contribution of switched memory B cells appears much greater. Can the authors clarify this discrepancy?

We agree that the differential cluster annotation used was confounding. In the previous version of the manuscript, we used the published SingleR package in R in previous Figure 1d and 1e. Cluster annotation is a result of the biological reference that was used. The SingleR algorithm was trained on human PBMCs. For consistency reasons, we now visualized all single cell datasets by the integrated Seurat v4 reference mapping algorithm. Readability of new Figure 1b and 1c should now be improved. Moreover, B cell subsets are now consistently defined throughout all downstream analyses.

6. Figure 1f, there appears to be a significant degree of background staining from both the IgG4 and IgG stains, which is something we often encounter in our clinical experience at least with IgG most commonly. According to some of the provided quantifications, it's not clear how true positive events were distinguished from non-specific background signal and I worry about misrepresentation of the true numbers of IgG4+ cells. Can the authors please clarify?

We have now included an automated assessment of IgG4+ cells and show histograms of IgG4 positivity and negativity by ROI staining intensity (new EV Figure S4) as means to validate semiautomatic quantification results provided in the previous version by automatic quantification as per default DAB parameters of the QuPath 0.2.3 software.

7. "Previous studies suggest that T follicular helper cells drive the class switch towards IgG4 via Interleukin-4 signaling." By earlier in vitro studies, it has been most strongly suggested that the combination of IL4 and IL10 is required for class-switching to IgG4. Did the authors observe a signal for IL10R transcription among the analyzed naïve B cells?

We thank the referee for this insight. This was indeed observed and is now included in Figure 1h.

8. "Interestingly, CD4+ T cell subsets demonstrated expression of PDCD1 and TCF7 consistent with a dysfunctional state." Follicular helper T cells have been exhaustively reported in the context of IgG4-RD expanded both in the blood and in the tissues. Did the authors observe co-transcription of PD1 and CXCR5 in the same cells and if so, were these cells also expressing relevant cytokines such as IL4, IL10, IL21? PD1 expressing CD4+ T cells have also more recently been described that are CXCR5- and termed peripheral helper T cells (TPH) by Deepak Rao and colleagues in the context of rheumatoid arthritis. Were the PD1 expressing CD4+ T cells observed here more fitting of this TPH phenotype, capable of helping B cells through CD40L but lacking CXCR5 expression?

We now include sub-analyses of CD4 T cell subsets in new EV Figure 5. Indeed, we observed TPH cells as defined by Rao et al. within the CD4 T cell subset (CD4 TEM / TCM /

naive) (now shown in EV Figure 5b,c). There are cells within the reference-mapped CD4 TCM compartment that includes TPH cells and T follicular helper cells that seem to be capable of helping CD40+ B cells by expressing CD40LG (Figure 1 below).

Figure 1. Average expression level of CD40 and CD40LG in T peripheral helper (TPH)-like cells (identified by reference mapping algorithm as follicular helper T cells) and B cells (identified by reference mapping algorithm as memory B cells).

9. CD4+CTLs have been described to be the most dominant CD4 T cell type at the site of disease as referenced in this manuscript, yet their contribution among the cells analyzed in the CSF is quite small. Can the authors discuss this discrepancy with the published tissue quantitative literature?

As previously hypothesized (answer to comment #2), the CSF pleocytosis is IgG4-RD-associated but not "IgG4-RD-lesion reflective". Activated B cells and plasmablasts additionally present antigens to CD4 CTLs in an MHC class II-dependent fashion leading to local TGF- β and IFN- γ production and ultimately fibrosis. It is tempting to speculate that the low number of plasmablasts / antigen presenting B cells in the CSF is responsible for overall low abundance of CD4 CTL. However, total abundance of CD4 CTL is increased in IgG4-RD CSF compared to CSF of MS patients and healthy controls (new Figure 2a) in accordance with published literature. (Mattoo, H. et al. *J. Allergy Clin. Immunol.* (2016); Maehara, T. et al. *Ann. Rheum. Dis.* (2017)). We have provided additional experimental evidence on a TCR level, that differential clonal expansion in the CSF and parenchymal lesions take place.

10. The CD4+CTL response reported in IgG4-RD has consistently shown marked clonal expansion. More recently, expanded CD8+CTLs in the blood of patients with IgG4-RD were also found to be highly clonally expanded (Perugino, Kaneko, Maehara, et al, 2021). However, here the authors find "a polyclonal TCR repertoire in both T cell compartments as determined by single-cell VDJ profiling." With all suggested B-T interactions, presumed antigen presentation and the reported literature on clonal expansion, how do the authors explain the diverse TCR repertoire observed here?

Please also refer to the answers to comments #2 and #9. We now provide longitudinal TCRB deep sequences of the IgG4-RD lesions (new Figure 3) and now discuss the results extensively in the revised manuscript.

11. The functional interaction analysis is very interesting. "direct T cell-T cell crosstalk between CD8 TEM and CD4+ memory cells" and "we found evidence of direct T cell - T cell

crosstalk via C-C chemokine ligand 5 (CCL5) on CD8 TEM binding to C-C chemokine receptor type 4/5 (CCR4/5)-expressing T cells that might facilitate recruitment of helper and cytotoxic CD4+ T cells that have been implied to mediate inflammation in IgG4-RD" - CCL5 made by the CD8s may have landed on its receptor on some other cell that wasn't among the cells subjected to scRNAseq. This is also location dependent. A lot of these are speculative assertions and should be presented as such. I would avoid using such declarative words like "direct" as this interaction is rather only suggested through indirect analyses.

We have now substantially revised the wording of these sections as they were indeed based solely on *in silico* interactions and predictions.

12. "We therefore propose a novel pivotal role for CD8 TEM in driving and sustaining IgG4-RD via distinct cell-cell interactions and chemoattraction of pathogenic B cells and cytotoxic CD4+ T cells." This is a very interesting idea and one absolutely worth investigating further. I do want to bring two references to the authors attention. One, the work by Perugino, Kaneko, Maehara et al published in JACI, mentioned above, the provides transcriptional phenotyping of CD4+CTLs and CD8+CTLs from the circulating cells from patients with IgG4-RD. CCL5 and CCR5 were two transcriptional signatures noted in the bulk effector CD4+CTL data. The 2nd, by Della-Torre et al, also published in JACI, described a potential direct role for B cells in driving fibrosis in this disease and CCL4 and CCL5 were explored extensively, both found to be secreted by pathologic B cells at the sites of disease and suggested as potential mediators to orchestrate the immune cell infiltrate in this disease. These ideas of cytokine mediated communication between relevant cell types in IgG4-RD has been a recurrent finding and these additional citations bolster the authors suggestions about the importance of these axes in this disease.

We apologize not having cited these highly relevant studies providing external evidence to our study and now cite these papers adequately.

Referee #3 (Remarks for Author):

Major

Parenchymal lesions of IgG4-RD in the central nervous system (CNS) are rare and the authors described a case of an inflammatory intracranial pseudotumor and reported single-cell sequencing of patient cerebrospinal fluid (CSF). This case is rare and informative in the field of CNS lesions in IgG4-RD. However, there are some questions to be resolved.

We thank the referee for this positive feedback.

Minor specific comments

1) To make a correct diagnosis of IgG4-RD, histopathological findings are the most important. However, interobserver concordance among pathologists in histopathological diagnosis is not necessarily satisfied. In the text, the authors mentioned characteristic histological features such as dense lymphoplasmacytic infiltrate, obliterative phlebitis and storiform fibrosis, but not full pictures lacking obliterative phlebitis and storiform fibrosis. Full pictures should be mentioned.

We agree and now show all histopathological findings in new EV Figure S2. Full images can be publicly accessed via the following accession code: <https://doi.org/10.6084/m9.figshare.14522397.v1>.

2) It has been established that IgG4-RD is TH2 dominated immune response, but not mentioned in anywhere results. This should be mentioned or discussed.

We have now assessed the phenotype of T cells in Figure 1g including Th2 signature cytokine expression and mention this observation in the result section.

26th May 2021

Dear Dr. Bunse,

Thank you for the submission of your revised manuscript to EMBO Molecular Medicine. I am pleased to inform you that we will be able to accept your manuscript pending the following final amendments:

- 1) Please make sure that it is clearly indicated that Figure 1F presents technical replicates and not biological replicates as suggested by the referee #2.
- 2) In the main manuscript file, please do the following:
 - Correct/answer the track changes suggested by our data editors by working from the attached/uploaded document.
 - Structure of the manuscript should be the following: Abstract, Introduction, Results, Discussion (or Results and Discussion), Materials and Methods, etc. Please check "Author Guidelines" for more information. <https://www.embopress.org/page/journal/17574684/authorguide#researcharticleguide>
 - Add up to 5 keywords.
 - Make sure that all special characters display well.
 - In M&M, provide the antibody dilutions that were used for each antibody.
 - In M&M, include that, in addition to the WMA Declaration of Helsinki, the experiments conformed to the principles set out in the Department of Health and Human Services Belmont Report.
 - In M&M, statistical paragraph should reflect all information that you have filled in the Authors Checklist, especially regarding randomization, blinding, replication etc.
 - Indicate in legends exact p -values, not a range, along with the statistical test used. To keep the figures "clear" some authors found providing an Appendix table Sx with all exact p -values preferable. You are welcome to do this if you want to.
 - Rename "Disclosure of Conflicts of Interest" to "Conflict of Interest".
 - In the reference list, citations should be listed in alphabetical order. Where there are more than 10 authors on a paper, 10 will be listed, followed by "et al.". Please check "Author Guidelines" for more information.

<https://www.embopress.org/page/journal/17574684/authorguide#referencesformat>

- In addition to the accession number please provide URL all deposited datasets. Please be aware that all datasets should be made freely available upon acceptance, without restriction. Use the following format to report the accession number of your data:

[data type]: [full name of the resource] [accession number/identifier] ([doi or URL or identifiers.org/DATABASE:ACCESSION])

Please check "Author Guidelines" for more information.

<https://www.embopress.org/page/journal/17574684/authorguide#availabilityofpublishedmaterial>

- 3) Funding: Please make sure that information about all sources of funding are complete in both our submission system and in the manuscript.
- 4) Source data: We appreciate making the imaging source data available via figshare. However, to be able to link a source data image with the respective figure we would like to encourage you to submit them to our submission system as one file per figure and name it "Source Data Figure 1" etc. (use a zip archive if multiple images need to be supplied for one Figure). Please check "Author Guidelines" for more information.

<https://www.embopress.org/page/journal/17574684/authorguide#sourcedata>

5) The Paper Explained: Please provide "The Paper Explained" and add it to the main manuscript text. Please check "Author Guidelines" for more information.

<https://www.embopress.org/page/journal/17574684/authorguide#researcharticleguide>

6) Synopsis:

- Synopsis text: Please submit synopsis text as a separate .doc file.

- Synopsis image: Please provide a striking image or visual abstract as a high-resolution jpeg file 550 px-wide x (250-400)-px high to illustrate your article.

- Please check your synopsis text and image, revise them if necessary and submit their final versions with your revised manuscript. Please be aware that in the proof stage minor corrections only are allowed (e.g., typos).

7) For more information: There is space at the end of each article to list relevant web links for further consultation by our readers. Could you identify some relevant ones and provide such information as well? Some examples are patient associations, relevant databases, OMIM/proteins/genes links, author's websites, etc...

8) As part of the EMBO Publications transparent editorial process initiative (see our Editorial at <http://embomolmed.embopress.org/content/2/9/329>), EMBO Molecular Medicine will publish online a Review Process File (RPF) to accompany accepted manuscripts. This file will be published in conjunction with your paper and will include the anonymous referee reports, your point-by-point response and all pertinent correspondence relating to the manuscript. Let us know whether you agree with the publication of the RPF and as here, if you want to remove or not any figures from it prior to publication. Please note that the Authors checklist will be published at the end of the RPF.

9) Please provide a point-by-point letter INCLUDING my comments as well as the reviewer's reports and your detailed responses (as Word file).

I look forward to reading a new revised version of your manuscript as soon as possible.

Yours sincerely,

Zeljko Durdevic

***** Reviewer's comments *****

Referee #1 (Comments on Novelty/Model System for Author):

Same as first round of Review

Referee #1 (Remarks for Author):

Compliments to the authors. They have done an impressive work in revising the work and adding a number of important novel experiments that clearly strengthen the message of the paper. I have no more comments to add but only a final curiosity: CD14 and CD16 monocytes/macrophages clouds seem to be significantly less abundant in the patient than controls. This might be an important pathological aspect to discuss but I understand that the paper is primarily focused on T cells. The authors might consider in the future to investigate this aspect on innate immunity which has been clearly neglected until now in the available literature.

Referee #2 (Comments on Novelty/Model System for Author):

All queries raised by this reviewer have been adequately addressed in the revision.

Referee #2 (Remarks for Author):

All queries raised by this reviewer have been adequately addressed in the revision. This manuscript is very interesting and adds as much to our understanding of the mechanisms that drive IgG4-related disease as could be obtained from a single patient. It's excellent work and I am very enthusiastic about it.

The only residual very minor issue I raise relates to Figure 1F in which technical replicates are plotted. This is confusing to a reader as the assumption will be that these are biologic replicates from different subjects. I would consider simply stating the value of CXCL13 found in this patient compared to controls in the text and not presenting the data in a figure format ("data not shown") or converting Figure 1F to a supplemental figure and including only the biologic replicates without statistical analysis.

Point-by-Point response

Re-Revision

Referee #1 (Comments on Novelty/Model System for Author):

Same as first round of Review

Referee #1 (Remarks for Author):

Compliments to the authors. They have done an impressive work in revising the work and adding a number of important novel experiments that clearly strengthen the message of the paper.

I have no more comments to add but only a final curiosity: CD14 and CD16 monocytes/macrophages clouds seem to be significantly less abundant in the patient than controls. This might be an important pathological aspect to discuss but I understand that the paper is primarily focused on T cells. The authors might consider in the future to investigate this aspect on innate immunity which has been clearly neglected until now in the available literature.

A sincere thanks for this comment.

Referee #2 (Comments on Novelty/Model System for Author):

All queries raised by this reviewer have been adequately addressed in the revision.

We thank the referee for this comment.

Referee #2 (Remarks for Author):

All queries raised by this reviewer have been adequately addressed in the revision. This manuscript is very interesting and adds as much to our understanding of the mechanisms that drive IgG4-related disease as could be obtained from a single patient. It's excellent work and I am very enthusiastic about it.

We are grateful for this comment.

The only residual very minor issue I raise relates to Figure 1F in which technical replicates are plotted. This is confusing to a reader as the assumption will be that these are biologic replicates from different subjects. I would consider simply stating the value of CXCL13 found in this patient compared to controls in the text and not presenting the data in a figure format ("data not shown") or converting Figure 1F to a supplemental figure and including only the biologic replicates without statistical analysis.

We agree with the reviewer. In Figure 1F experimental repeats with technical replicates have been shown. We therefore removed statistical analyses.

Additional editorial comments:

1) Please make sure that it is clearly indicated that Figure 1F presents technical replicates and not biological replicates as suggested by the referee #2.

This question has been addressed.

2) In the main manuscript file, please do the following:

- Correct/answer the track changes suggested by our data editors by working from the attached/uploaded document.

This question has been addressed.

- Structure of the manuscript should be the following: Abstract, Introduction, Results, Discussion (or Results and Discussion), Materials and Methods, etc. Please check "Author Guidelines" for more information.

<https://www.embopress.org/page/journal/17574684/authorguide#researcharticleguide>

- Add up to 5 keywords.

- Make sure that all special characters display well.

- In M&M, provide the antibody dilutions that were used for each antibody.

- In M&M, include that, in addition to the WMA Declaration of Helsinki, the experiments conformed to the principles set out in the Department of Health and Human Services Belmont Report.

This question has been addressed.

- In M&M, statistical paragraph should reflect all information that you have filled in the Authors Checklist, especially regarding randomization, blinding, replication etc.

This question has been addressed.

- Indicate in legends exact p - values, not a range, along with the statistical test used. To keep the figures "clear" some authors found providing an Appendix table Sx with all exact p - values preferable. You are welcome to do this if you want to.

This question has been addressed.

- Rename "Disclosure of Conflicts of Interest" to "Conflict of Interest".

This question has been addressed.

- In the reference list, citations should be listed in alphabetical order. Where there are more than 10 authors on a paper, 10 will be listed, followed by "et al.". Please check "Author Guidelines" for more information.

<https://www.embopress.org/page/journal/17574684/authorguide#referencesformat>

This question has been addressed.

- In addition to the accession number please provide URL all deposited datasets. Please be aware that all datasets should be made freely available upon acceptance, without restriction. Use the following format to report the accession number of your data:

[data type]: [full name of the resource] [accession number/identifier] ([doi or URL or identifiers.org/DATABASE:ACCESSION])

Please check "Author Guidelines" for more information.

<https://www.embopress.org/page/journal/17574684/authorguide#availabilityofpublishedmaterial>

This question has been addressed.

3) Funding: Please make sure that information about all sources of funding are complete in both our submission system and in the manuscript.

This question has been addressed.

4) Source data: We appreciate making the imaging source data available via figshare. However, to be able to link a source data image with the respective figure we would like to encourage you to submit them to our submission system as one file per figure and name it "Source Data Figure 1" etc. (use a zip archive if multiple images need to be supplied for one Figure). Please check "Author Guidelines" for more information.

<https://www.embopress.org/page/journal/17574684/authorguide#sourcedata>

This question has been addressed.

5) The Paper Explained: Please provide "The Paper Explained" and add it to the main manuscript text. Please check "Author Guidelines" for more information.

<https://www.embopress.org/page/journal/17574684/authorguide#researcharticleguide>

This question has been addressed.

6) Synopsis:

- Synopsis text: Please submit synopsis text as a separate .doc file.

- Synopsis image: Please provide a striking image or visual abstract as a high-resolution jpeg file 550 px-wide x (250-400)-px high to illustrate your article.

- Please check your synopsis text and image, revise them if necessary and submit their final versions with your revised manuscript. Please be aware that in the proof stage minor corrections only are allowed (e.g., typos).

This question has been addressed.

7) For more information: There is space at the end of each article to list relevant web links for further consultation by our readers. Could you identify some relevant ones and provide such information as well? Some examples are patient associations, relevant databases, OMIM/proteins/genes links, author's websites, etc...

This question has been addressed.

8) As part of the EMBO Publications transparent editorial process initiative (see our Editorial at <http://embomolmed.embopress.org/content/2/9/329>), EMBO Molecular Medicine will publish online a Review Process File (RPF) to accompany accepted manuscripts. This file will be published in conjunction with your paper and will include the anonymous referee reports, your point-by-point response and all pertinent correspondence relating to the manuscript. Let us know whether you agree with the publication of the RPF and as here, if you want to remove or not any figures from it prior to publication. Please note that the Authors checklist will be published at the end of the RPF.

We agree with the publication of the RPF.

9) Please provide a point-by-point letter INCLUDING my comments as well as the reviewer's reports and your detailed responses (as Word file).

This question has been addressed.

4th Jun 2021

Dear Dr. Bunse,

We are pleased to inform you that your manuscript is accepted for publication and is now being sent to our publisher to be included in the next available issue of EMBO Molecular Medicine.

Corresponding Author Name: Lukas Bunse, MD, PhD

Manuscript Number: EMM-2021-13953-V2